# Silent Tears of Midwives: ‘I Want Every Mother Who Gives Birth to Have Her Baby Alive’—A Narrative Inquiry of Midwives Experiences of Very Early Neonatal Death from Tanzania

**DOI:** 10.3390/children10040705

**Published:** 2023-04-10

**Authors:** Jan Becker, Chase Becker, Rachel Abeysekera, James Moir, Marion Gray, Meshack Shimwela, Florin Oprescu

**Affiliations:** 1Midwife Vision Global, Uhuru Street, Dar es Salaam 12101, Tanzania; 2Medical School, University of Nicosia in Partnership with St George’s University of London, Makedonitissis 46, Nicosia 2417, Cyprus; 3Centre for Health Research, School of Health and Wellbeing, Faculty of Health, Engineering and Sciences, University of the Southern Queensland, Toowoomba, QLD 4300, Australia; 4Temeke Regional Referral Hospital, Dar es Salaam 15101, Tanzania; 5Public Health, University of Sunshine Coast, Sippy Downs, QLD 4556, Australia

**Keywords:** very early neonatal death, resilience, stories, self-efficacy, sadness, active agency, global health, low-income settings

## Abstract

Background: Midwives working in settings with limited clinical resources experience high rates of very early neonatal deaths. Midwives manage the impact of this grief and trauma almost daily, which may affect patient care and their own well-being. Research Aims: To explore how midwives are impacted by and cope with high rates of very early neonatal deaths. To document midwives’ insights and local solutions that may reduce very early neonatal deaths in limited resource settings. To document the stories of midwives in order to create awareness and garner support for midwives and their critical work in low resource settings. Methods: Narrative inquiry utilizing semi structured interviews. Twenty-one midwives with at least six months experience who had experienced or witnessed very early neonatal death were interviewed. Data were audio recorded and transcribed, and reflexive thematic analysis of transcripts was conducted. Results and Discussion: Three themes were identified: (1) deep sadness resulting from very early neonatal deaths leading to internal struggles; (2) use of spirituality, including prayer and occasional beliefs that unexplainable deaths were ‘God’s plan’; and (3) development of resilience by seeking solutions, educating themselves, taking accountability and guiding mothers. Participating midwives noted that inadequate staff and high caseloads with limited basic supplies hindered their clinical practice. Participants articulated that they concentrated on active solutions to save babies during labour, such as vigilant foetal rate heart monitoring and partogram. Further, reduction and prevention of very early neonatal death is a complex problem requiring multidisciplinary teams and woman-centred care approaches to address issues contributing to the health of mothers and their new-borns. Conclusions: Midwives’ narratives highlighted ways of coping with grief and deep sadness, through prayer, and further education of both mothers and fellow colleagues to achieve better antenatal and intrapartum care and outcomes. This study gave midwives an opportunity for their voices to be heard and to generate solutions or insights that can be shared with colleagues in similar low-resource settings.

## 1. Introduction

In low-resource settings, midwives are burdened by high birth rates and low staff-to-patient ratios [1]. On an almost daily basis, they face high rates of very early neonatal death (VEND), which involves the death of a newborn within the first 24 h after birth [2]. For example, in Tanzania, midwives are primarily responsible for managing labour and birth, including complex maternal cases and neonatal resuscitations [3]. The primary causes of VEND include prematurity and birth asphyxia [4]. An understanding of the impact of VEND and the ways in which midwives cope with this phenomenon may enable us to develop ways of supporting midwives’ psychological well-being and intervention strategies with the aim of enhancing their resilience and health care delivery under difficult conditions [5].

Previous research has investigated midwives’ experiences with neonatal death in high-income countries (HICs) [6]. However, these experiences are likely to differ in limited resource settings, such as Tanzania, and other low-income countries (LICs) [5]. Prevention and reduction of these very early deaths is a complex problem requiring a multifactorial approach to address preconception, antenatal and high-risk antenatal care for mothers (for example, syphilis, anaemia, malaria, diabetes, and eclampsia), availability of transport to higher care, intrapartum monitoring and delivery, which can affect the outcome of neonates [7]. The work with labouring mothers can make midwives vulnerable to psychological stressors, which may ultimately lead to their departure from the profession, leading to an even more reduced and unskilled workforce [8].

Midwives in Tanzania face emotional stress and professional challenges due to the country’s very high neonatal mortality rates, which may be worse than previous research has suggested [9]. Worldwide, approximately 1 million neonates die within the first 24 h after birth [10]. Approximately 99% of these deaths occur in low- and middle-income countries (LMICs) [11]. Tanzania has over 40,000 neonatal deaths per annum, with the leading causes of death being preterm births, sepsis, and birth asphyxia [12].

Multiple factors can be considered when a midwife experiences the death of a newborn, including the need to counsel the grieving mother, the midwife’s own grief, compassion fatigue, posttraumatic stress, and other influences on the midwives’ psychological well-being, burnout, along with the subsequent impact of this phenomenon on the midwife’s psychological well-being [13,14]. Midwives in sub-Saharan Africa rarely have the time to grieve due to the almost continuous overwhelming patient workloads [15]. While labour wards in HICs are not expected to experience neonatal deaths routinely, this is a frequent occurrence in Tanzania [6]. Due to the high rates of still births and VEND in Tanzania, the research hospital has an area in the labour ward labelled ‘dead baby bench’ upon which babies that have died are wrapped in kangas (Tanzania cotton cloth culturally significant on the eastern coast of Africa, often given as a gift) to await transport to the morgue. Such a bench would be unimaginable in the labour ward of a HIC.

Currently, there is a paucity of research capturing midwives’ narratives on coping, along with their insights and solutions for reducing preventable VEND in Tanzania. This research project collected experiences and solutions for midwives that may reduce these deaths as well as insights in what coping mechanisms may be useful [16]. Narratives are a powerful way for midwives to share their stories [17] and acknowledge their disenfranchised grief [18]. This research is in line with the message of the 2016 WHO report “Midwives Voices Midwives Realities” [9].

### Research Aims

To explore how midwives are impacted by and cope with high rates of very early neonatal deaths. To document midwives’ insights and local solutions that may reduce very early neonatal deaths in limited resource settings. To document the stories of midwives in order to create awareness and garner support for midwives in their critical work in low resource settings.

## 2. Materials and Methods

### 2.1. Study Design

This study applied a qualitative narrative inquiry design that utilized the consolidated criteria for reporting qualitative studies (COREQ32) checklist (Appendix A). Semi structured interviews were audio recorded using a digital device. This method was selected because it provided insight into midwives’ feelings and experiences.

### 2.2. Study Setting and Context

This study was conducted at a referral hospital located in the Ilala District of Dar es Salam, Tanzania, where midwives manage many births (between 30 and 100 per day). For example, in this hospital there were 13,618 births recorded in 2019, most of them assisted by midwives only as doctors in the hospital manage complex emergency cases and surgery. Dates: January–June 2019 saw 256 Neonatal Deaths (Includes 22 Intrauterine Fetal Death (IUFD), Fresh Still Birth & Macerated Still Birth). 206 (Labour ward) and 50 (Neonatal Unit).

All staff in the study confirmed they have experienced VEND—and many stated they had seen many every week. Excluding the IUFD neonatal deaths in labour ward is on average one neonate dying per day. Further the ward is open plan with mothers less than one metre away from each other and staff shortages meant midwives were involved in or witnessing VEND often.

### 2.3. Recruitment and Participants

Midwives who had worked a minimum of six months in the labour ward or the neonatal unit were included in the study. These participants all confirmed they had managed or witnessed VEND, ensuring that we could obtain rich data. Participants were recruited by displaying a poster and distributing printed hand-outs until data saturation was reached [19]. Participation was voluntary, and the midwives were not paid for their participation in the study.

### 2.4. Ethics

Ethics approval was obtained from the National Institute for Medical Research, United Republic of Tanzania (NIMR/HQ/R.8a/Vol. IX/2828) and the Human Research Ethics Committee at the University of the Sunshine Coast, Queensland, Australia (S191339) prior to recruitment. Verbal consent was obtained at the start of online and written communications. Written participant consent was obtained during the face-to-face interviews. Confidentiality was ensured by using a private room at the hospital. Data were anonymized before being disseminated to other researchers. Participants could withdraw from the study at any time without penalty.

### 2.5. Data Collection

Interviews with 21 midwives were conducted in English (77%) and Kiswahili (23%). The researchers (JB, JM, and CB) had worked/volunteered in this location for more than 7 years, which resulted in the development of rapport and trust with the potential participants. The SARS-CoV-2 (COVID-19) global pandemic impacted in-person data collection; hence, a more flexible data collection strategy was used. Midwives appeared committed to the task of telling their stories regardless of the platform used for the interview. Flexibility on data collection using various online and email interviews has been adopted by other studies [20] which have generated rich data.

Eight in-person interviews of 45–60 min were conducted, with the remaining thirteen online interviews lasting 25–35 min. Six interviews were conducted via written narratives, and seven interviews were conducted via internet-based applications (e.g., WhatsApp^©^) [21].

An initial two-question interview guide was used (‘How do you cope with so many very early neonatal death perinatal deaths?’; ‘What things do you do to help cope with dying babies?’). These questions were adjusted after the initial 2–3 interviews to prompt further clarification and elaboration. Verbal probes were used to encourage the participants to share specific strategies, feelings, and actions. Emails and WhatsApp communications were used to follow up with participants within 12 weeks [22]. Field notes and reflexive diary entries captured any nuances, tones, and silences during and after the interviews. Data collection ceased after achieving data saturation, as no new significant information was forthcoming [19].

### 2.6. Data Analysis

All audio-recorded interviews were transcribed verbatim. Written narratives were translated professionally and checked for accuracy. Participants were deidentified by an assigned number, and identifiable details were stored in password-protected files. No themes were identified in advance; instead, the analysis was conducted in an active, creative, inductive manner, such that concepts were primarily derived and themes were conceptualized from the raw data categorization by a researcher. Researchers, JB, CB and RA independently identified patterns, clusters of similar meaning. All joined the robust discussion several times with final themes and quotes agreed upon to enhance rigor and further reduce bias. The analytical framework of Braun et al. [23] guided the stages of the interpretive reflexive thematic analysis process (see Table 1 for detail).

Using hard copies, highlighters and pencils initially, followed by each researcher independently placing quotes and themes into a word document for group discussion. The final themes were selected to best express the meanings presented in the participants stories [24]. This was not a linear process; rather, researchers returned purposefully, and repeatedly to engage critically with the data to ensure that themes captured the depth and breadth of the midwives’ narratives. Analytical discussions were conducted until a consensus was reached by all authors.

The themes were derived from the data—not assigned to an already labelled theme. The analysis was a long, detailed, and robust process—the thematic development with a thorough back and forth process with much discussion, with each researcher always keeping the dataset in the forefront of the process.

The process of what we learned, derived, and analysed from the midwives’ narratives focused on similar narratives that created the ‘themes’ (categories).

The themes identified focus attention on the experiences of midwives and the need to address sadness resulting from VENDS via spiritual practices/beliefs and solution generation/adoption.

**Table 1 children-10-00705-t001:** Stages of Braun and Clarke’s reflexive thematic analysis [25] utilized to analyse the data collected from interviews with the midwives.

Stage	Stage Details	Procedures in Each Stage
1	Familiarization with the data	Researchers both read and reread transcripts and reviewed audio files to immerse themselves in the interviews. During this process of familiarization, notes were made concerning initially relevant and noteworthy interpretations. Researchers began searching for patterns and meaning.
2	Generating initial themes from the data	Researchers identified interesting features of the data, which were collated into a table of evolving topics, including relevant quotations with notations labelling potential themes. This analysis, alongside the research question and aims, allowed semantic themes to be conceptualized using underlying patterns, words, and impressions. The data were labelled and organized into a table.
3	Searching for meaningful patterns and conceptualizing themes	Meaningful patterns were identified as the researchers actively sorted highlighted words and quoted phrases into the themes. Quotations relevant to each potential theme were further collated into tables by selecting more definitive quotations. Researchers identified the meaning between the initial themes and words.
4	Revising themes	Potential themes were organized into one table based on the individual notes. Collaborative dialogues were held. Some themes were merged, while others were removed or separated into two themes. The table was expanded by using mapping to connect quotations and comments. Researchers identified the compelling stories and defined links among the various themes in an active, robust process that involved returning to the original interview data many times to ensure that the narratives resonated authentically.
5	Themes defined and established	An ongoing analysis was conducted focusing on the ‘core’ of each theme, all of which were given clearer, more concise names to ensure that the data extracted were well represented. Each theme was discussed at length as the final themes were developed; the data sets were constantly revisited during this process. The data analysis was a recursive process.
6	Report finalized	Themes were finalized. Specific quotations and extracts were selected to highlight the themes most effectively. The written report went beyond a simple description by presenting interesting accounts of the stories ‘within and across themes’ [26].

### 2.7. Rigor

Participants of different ages and with different levels of clinical experience were sought to optimize the representation of midwives in this study as well as to confirm its credibility and support its transferability [27]. Rigor was supported by the COREQ32 checklist criteria [28]. Narrative inquiry was best suited to elicit rich stories from midwives, whose voices have been rarely heard. An interview guide was utilized, and interviews were conducted by the same researcher to enhance reliability [29].

To address the potential influence of preconceptions and bias, a collaborative acknowledgement was made by highlighting the clinical experiences at the research site alongside perceived biases and assumptions by the researchers JB, CB, RA, MS and JM. Prolonged engagement and “fitting in” by JB, CB and JM were achieved by immersion in the cultural and clinical context for more than 7 years, which supported reflexivity and encouraged participants to share their stories more openly [30]. All authors contributed to the diversity of the research in terms of professional and academic skills, which enhanced the final reflexive thematic considerations of this research and thus its trustworthiness [31]. The research group conducted peer review sessions to ensure the trustworthiness of the midwives’ stories [31].

## 3. Results

Twenty-one midwives (1 male and 20 females) aged between 26 and 54 years with clinical experience ranging from 1 to 25 years (median 10.5 years) were interviewed. Many of the participants had experienced hundreds of cases of VEND. Two interviews were complicated by internet connectivity; thus, these midwives were interviewed twice. Data were collected between July 2019 and February 2021.

### 3.1. Themes

Based on the data of the midwives’ narratives, three major themes were identified.

Deep sadness and internal struggleSpirituality, including the use of prayer and occasional beliefs that unexplainable deaths were ‘god’s plan’; andDeveloping resilience by seeking solutions, educating themselves, taking accountability and guiding mothers

Narratives extended across all three themes, suggesting multifaceted ways of coping with and making sense of neonatal deaths and demonstrating the emotional connection between the midwife (the professional), who learned from the neonatal deaths while endeavouring to achieve better outcomes for the babies (humanness), and the fragility of the midwife [6].

### 3.2. Theme 1 Deep Sadness and Internal Struggle

Deep sadness was a recurring theme which was universally discussed first in all interviews. One midwife described sadness as follows:

‘*It is painful to have many perinatal deaths and difficult to cope but I try my best to reduce them*’.Mkunga 02. Mkunga means midwife in Kiswahili.

Sadness was followed closely by internal struggles resulting in deliberation of the question “what if”; could the baby’s death have been prevented, and what could have been done differently [32]? This expression of contrition focused on questioning whether the midwife’s clinical skills were lacking or whether different interventions could have produced a different result; such thoughts seemed to add to the burden of grief the participants faced.

Furthermore, midwives discussed their inability to avoid pain as well as the lack of emotional support or training and skills necessary to cope with the situation themselves or to give compassionate, appropriate counselling to the mother. These struggles with blame and the tendency to question the clinical sequalae during labour and birth were evident in multiple responses.

‘*It is difficult to cope when a baby dies. It is so painful. Nowhere to escape.**If the baby is delivered with a low score and resuscitation is done in the golden minute and dies, I ask myself,* ‘*Maybe my lack of instruments, poor management, or delayed treatment*’.Mkunga 19


*One midwife’s story was particularly evocative. In the interview, she spoke very quietly as she recounted the story of one of the hundreds of VENDs that she said she had experienced.*


‘*A mama came from a peripheral hospital… It takes a long time to come to the hospital due to transport; there are no ambulances. I tried to resuscitate… I did it for almost 20 min, but nothing; it was very painful, and that mother, she was gravida three, para zero*Gravida 3, Para 0 indicates that this mother has no living babies despite three pregnancies.*Therefore, to me, it was very, very painful for her, and I felt the pain bad that day*’Mkunga 04

### 3.3. Theme 2 Spirituality, including the Use of Prayer and Occasional Beliefs That Unexplainable Deaths Were ‘God’s Plan’

One-third of the participants (33%) discussed using prayer to cope. Connecting to God was discussed by most participants. One midwife summarised this concisely in the following statement:

‘*I pray to overcome the sadness’.*
Mkunga 08 

When attempting to make sense of the death, some midwives continued to consider, rationalize, and question the event and subsequently returned to God to help them.

‘*When a baby dies, I’m emotionally hurt. I spend days thinking about the event.**Sometimes, I say to myself that maybe it was God*’*s plan, but sometimes, I keep asking myself if it was our fault as health workers, failing to intervene.**In the end, I pray and ask God not to let a baby die in my hands*’.Mkunga 16

One midwife noted that she had nightmares following VENDs.

‘*Two weeks back, I had a bad nightmare after coming from the meetings we used to discuss very early neonatal death, fresh stillbirth, and maternal death. When I sleep at night, I feel like a baby is chasing me, and I am running and running…*’Mkunga 02

Midwives displayed an authentic acknowledgement of their sorrowful feelings that were partially mitigated by spiritual connectivity.

### 3.4. Theme 3 Developing Resilience by Seeking Solutions, Educating Themselves, Taking Accountability and Guiding Mothers

The ward often has only 2 midwives on night duty. All mothers stay in one room, electricity is unreliable, there is no soap, the supply of basic medications is limited, and a lack of staff hampers care significantly [33]. Resilience is a needed element under such circumstances. One midwife expressed her resilience as follows:

‘*First, I stop doing anything at that moment, and I sit down to relax and relieve the panic that I have at that moment according to the situation (the baby dying).*
*Second, I tell myself that it is okay because anything can happen in the medical field, death is not the end, and failure to save the baby does not mean that I will fail to save another baby, and I will look at it as a lesson to save other babies.*
*I give myself the belief that I can save other babies*’.Mkunga 16 

Theme 3 highlighted the midwives’ inclination towards active solution seeking rather than passive observation; it was also associated with an emphasis on motivation regarding learning from mistakes and being part of the change or solutions for the better.

‘*We are doing a perinatal death review to find the cause of death and teach ways of preventing death*’.Mkunga 05

Midwives reported frustration towards colleagues who did not appear to care and suggested that the solution was to transfer these individuals to a different ward, as highlighted in the following quotations:

‘*Some of the deaths that occur are caused by some workers who do not care*’.Mkunga’ 01

‘*If everyone was to give her skills to the mother, if you do not know, ask someone who has skill, get the skill, and save the baby.**If not satisfied with the labour ward, go to another ward*’. Mkunga 06

Midwives mentioned focusing on education (52%) and using other strategies to avoid VEND (see Table 2). Their empathy extended to practical actions, such as education regarding pregnancy spacing, the inadvisability of using local herbs to accelerate labour, the importance of visiting the hospital sooner, the need to attend antenatal clinics to identify risks, and the essential nature of vigilant monitoring during labour.

While providing such education within minutes of a baby’s death is likely not appropriate, there is limited time to counsel and provide emotional support due to the high volume of mothers in labour. This emphasis on prevention reflected the midwives’ individual self-efficacy regarding their contributions to the task of preventing VEND via collective self-efficacy (see Table 2).

## 4. Discussion

This study sought to explore and document how midwives cope with high rates of very early neonatal deaths and gain midwives’ insights and solutions for this issue.

The insights and solutions conveyed by the midwives can have clinical impact and usefulness simply by allowing the stories to be heard, opening dialogues with management to adopt a team problem solving approach for example, streamline labour ward caseloads, utilization of partograms and support neonatal resuscitation training.

Other studies have documented that the primary impact of neonatal death on midwives is trauma, which creates burn-out and compassion fatigue [34,35,36,37,38]. VEND is traumatic due to the circumstances in which it occurs [39]. One study highlighted the importance of validating nurses’ feelings regarding traumatic situations to help them grow from these situations and enhance their future practices.

Other studies have indicated that the impact of neonatal death is burnout, which is severe enough for midwives to consider leaving the profession [14,40]. The results of this research project suggest that midwives cope more effectively with burnout when they seek social, spiritual, familial or collegial support [36], a claim that resonated with many midwives in this study, as they sought solace by relying on prayer, family and peers [15]. Several midwives drew strength from helping women physically, if not psychologically, while noting that leaving a grieving mother on her own triggered feelings of guilt and affected their mental health [37,41].

Some studies of midwives in HIC have shown that they have a number of resources and a large counselling network that could assist midwives in LIC if made available [6,40,42,43]. Kain’s [44] study defined grief as a “pervasive, highly individualized, dynamic process”. Tanzanian midwives shared stories that emphasized the fact that grief is not linear or localized but rather temporal, drawing on individualized experiences in both the present and past while simultaneously discovering new possible outcomes [13,18,45,46]. Such resilience is potent; based on the midwives’ accounts of haunting sadness, some have found a way to continue in a profession that is demanding in terms of the “self” [47].

Prayer has been utilized by midwives as a coping strategy, giving pause while providing meaning by connecting the internal questioning of one’s own competency with an external manifestation in terms of spirituality that allows midwives to overcome their own negative responses [46,48,49]. Other studies conducted in sub-Saharan Africa have found that despite systemic failures, overwhelming workloads, and poor resources, midwives have elected to remain and “just keep going” [5,15]. Sharing their stories may allow midwives to develop a supportive network; by hearing others’ narratives, they can realize that they are not alone [50]. This benefit is useful to mention because “stories bring meaning into our lives, convey values and emotions” [51] and can reveal the differences and similarities among people’s experiences. Through reflection, these stories can help develop resilience [51].

A lack of training on managing a mother’s grief when a baby dies is highlighted in LICs, which creates opportunities for in-service and training that fosters high self-efficacy; this could have a positive impact on managing performance [52,53]. Furthermore, grief counselling and better communication skills may provide more coping strategies, as one study found with healthcare workers experiencing less anxiety, increased confidence, and better relationship skills [54]. A supportive organizational ethos and change in culture from ‘blame’ to support would assist midwives in adjusting, minimizing psychological distress, and being better equipped to provide quality care [55,56].

Midwives’ stories can be impactful with regard to supporting positive changes [57] and resilience. Furthermore, recounting stories of hardship and difficult situations, such as experiences with VEND, can be a cathartic and therapeutic experience for research participants [51]. Midwives expressed a variety of emotions, including anger, stress, and demoralization; they all articulated feelings of sadness as well as the psychological consequences of self-reproach, which have been reported in other studies conducted in sub-Saharan Africa [15,58,59]. Furthermore, the physical exhaustion, tiredness and nightmares experienced by midwives have been the focus of other studies [16,60]. Midwives can be at risk of ‘burnout syndrome’, which can impact compassionate quality care in LICs [61].

Resilience can evoke multiple behaviours and activities rather than a set of characteristics [5]. Examples of resilience include the establishment of step-by-step processes towards specific goals, positive interactions with colleagues, and a largely optimistic and self-confident outlook, all of which were expressed by the midwives [5]. Instead of falling into introspective pity or self-effacing despair, these midwives rallied by becoming agents of change rather than pawns within a system that was seen most acutely in Ghana whereby the midwives and doctors stated perinatal death was ‘*part of the job*’ [15].

Resilient behaviours and attitudes were evidenced by the midwives’ focus on providing better care through education and better management of both normal and complex labour, which highlighted their mindset of being an agent of change despite the situation [62]. Well-managed debriefs that allow self-reflection may assist midwives in communicating news to the mother that her baby has died [63,64]. Additionally, midwife-led continuity of care has shown promise in Ethiopia, improving maternal and neonatal outcomes during antenatal, labour and birth [65]. Based on the literature and the findings of this study, resilience can be defined in terms of adaptation to difficult ongoing challenges and traumatic events with mental, emotional, and behavioural elasticity [47]. Participating midwives embraced the opportunity to share their stories and exhibited high levels of perceived collective self-efficacy as they navigated through an active agency of change leading to ‘resilience to adversity’ [66,67].

Underlying many of the collated narratives was the midwives’ frustration due to the failure to receive sufficient resources, their high caseloads, and systemic failures, which are commonplace in Tanzania [16,68,69]. The solutions for addressing high numbers of VEND are not simple, nor do they rest with midwives alone. A multidisciplinary team addressing multiple factors including socioeconomic, tribal traditions such as herbs to accelerate labour, health system capacity, and overall nutrition and health quality of individual mothers, family planning, prenatal folic acid, quality antenatal care, early transport to hospital and timely interventions may be employed when pregnancy and intrapartum problems arise [70]. The insights and solutions conveyed by the midwives can have clinical impact and usefulness simply by allowing the stories to be heard and opening dialogues with management to adopt a team problem solving approach. For example, streamlining labour ward caseloads, utilization of partograms and support for neonatal resuscitation training.

Many participants remained optimistic despite systemic difficulties; they learned lessons and built the self-confidence necessary to believe that babies can be saved in the future. Some examples of the midwives’ diverse positive adaptative behaviours and strategies were focused maternal education, concise clinical skills training, purposeful foetal heart monitoring, and the initiation of resuscitation at the bedside through the Champion program that advocates mentored hands-on-training during real live neonatal resuscitation [71]. Furthermore, midwives found ways of utilizing expired/single use equipment by resterilizing kiwi cups as many as 25 times, thereby saving babies’ lives.

Based on the aforementioned body of research and the current status of VEND in Tanzania, the task of addressing the ongoing high rates of VEND and the consequences of that situation seems to be an insurmountable problem [15,39]. Several cultural, political, economic, and social factors must be considered to understand the additional factors that complicate the appraisal of this situation. In places where birthing facilities have “dead baby” benches and neonatal deaths are a common occurrence, how do midwives cope?

Worldwide, midwives’ value and impact cannot be underestimated; even in HICs, the training and support of midwives is vital for maintaining a sustainable and resilient workforce [72]. Most midwives in Tanzania arguably do not have the liberty to choose to leave their professions, especially if the alternative is poverty [16]. It can be assumed that many midwives in Tanzania *may wish to leave* the profession due to severe psychological stressors but continue to work to provide for their families and sustain their lives [73].

Solutions for VEND are not simple; VEND remains a complex interplay between socioeconomic factors, maternal education and health, and tribal and cultural traditions. With the management of safe labour and birth beginning many months before with prenatal and antenatal care and managing labour and delivery with attentiveness. A multidisciplinary approach addressing access to higher care, available transport, and timely actions in light of obstetric complications, all contribute positively to reducing VEND. Importantly, research project such as this one can be a conduit to validate the day-to-day struggles of midwives in low resource areas and provide them and their colleagues with a sense of solidarity and hope.

## 5. Limitations

This study was conducted in part during the COVID-19 epidemic. Several methods of interviewing were used, including phone-based interviews, in which it was difficult to observe visual cues and emotions [74]. In terms of self-reporting bias, participants may have focused on positive interventions, such as education, as more favourable experiences instead of discussing their feelings of inadequacy as a result of social desirability bias. Additionally, the sample may not have been representative of the experiences of other midwives in rural Tanzania. Finally, the insider perspectives of two researchers (JB and CB) may have altered the participants’ narratives due to the effort to be ‘loyal’, and further interviews may be shaped by the insider’s empathy and experience. These limitations are important to mention for the benefit of future research projects.

## 6. Conclusions

In low-income countries, midwives are at the forefront of the phenomenon of VEND and may provide insight for potential solutions. Midwives’ narratives showed that coping with deep-seated sadness resulting from frequent experiences with VEND was in part eased by spirituality and prayer, with a belief that neonatal deaths were the result of ‘gods plan’. As agents of change, midwives focused on providing further education of both mothers and fellow colleagues to guide better antenatal and intrapartum care and outcomes. Midwives articulated that they concentrated on active solutions to save babies during labour, such as vigilant foetal heart monitoring and partogram, being prepared and conducting skilled neonatal resuscitation, rather than on the lack of staff, basic supplies, and overwhelming caseloads. It is important to note the high levels of resilience exhibited by some of these midwives working in low-resource areas despite abject grief and demoralization and their willingness to learn to impact preventable very early neonatal deaths. Despite being agents of change, many midwives were fatigued by the continual loss of babies.

Access to debriefing rather than blame, as well as psychological counselling skills, may encourage midwives to remain in the profession. Additionally, systemic workable solutions, teamwork, and the development of collective efficacy may contribute to empowering midwives in the quest to save babies. Encouraging training with incentives (financial or in-kind) to promote longevity and motivation may be worthwhile.

Midwives in Tanzania have been criticized in relation to poor neonatal outcomes. Few researchers have interviewed midwives to gain insights and identify possible solutions to inform practice and reduce very early neonatal deaths. The midwives’ narratives showed that they are forwards-looking in their actions and committed to learning and changing. While they experience deep sadness, their spirituality gives them the courage to overcome stagnation. Their perseverance to obtain better outcomes is a testament to their professional resilience.

Every midwife is worthy of being heard; we must listen to their stories to support our colleagues in their grief, in their solution seeking endeavours and in their future clinical practice.

## Figures and Tables

**Table 2 children-10-00705-t002:** Quotations from midwives focused on education and care strategies used to prevent Very Early Neonatal Death.

Midwife	Quoted Narratives
Mkunga	
02	To help midwives provide good care, I always teach them good practices for delivery, such as helping babies breathe.
03	Counsel the mother and next of kin and educate them on the cause of the death so that next time, they can take precautions. We advise the use of family planning. Plans to reduce the number of perinatal deaths: all pregnant mothers in labour should not exceed 8 h before delivery, close monitoring of partograph, early detection of any complications and action to be taken accordingly.
04	For perinatal death, in order for the number to be low, it is necessary to make sure that every nurse working in the labour ward should know how to resuscitate the baby by using Golden Minute and early diagnosis of foetal distress. Early decision on delivery to make sure that no woman stays in labour for more than 8 h according to dilation of the cervix. Additionally, to monitor maternal and foetal well-being using a partograph.
05	Educate family about the risk factors that cause perinatal death.
06	By giving a mother health education about attending or visiting a clinic. Make sure all pregnant mothers should not stay more than 12 h. This can prevent death.
07	By educating pregnant women about health as well as recognizing the signs of danger that can lead to complications and deaths. Provide maternal and child health education.
08	Things I can do to deal with dying babies. First, I need to be well equipped and prepare the mother before giving birth.
09	By educating and motivating each other, health workers and educating mothers as well.
10	When the mother arrives at the hospital to give birth, I admit her and check her to see the health of the mother and the unborn child. Check the status of risk indicators and provide timely care. Check the mother every four hours to know her progress. Measure the baby’s heart rate every half-hour. Provide health education to all mothers regarding risk indicators or tips.
20	Close monitoring of the partograph during labour. Educating the mother on danger signs during pregnancy. Early attendance at antenatal clinics and screening to exclude any complications from pregnancy. Application to help babies breathe as soon as possible after delivery for very sick babies or asphyxiated babies. Early admission of babies with a risk of infection to a neonatal unit. Advising the management to buy equipment used for resuscitation.
21	Provide health education to health workers in the proper way to maintain the pregnancy of the mother. A good attitude and preparation for midwives on providing healthcare. Fix in your mind that we can prevent neonatal death. We can work for a good attitude and cooperation.

## Data Availability

The data that support the findings of this study are available from the National Institute for Medical Research, Tanzania; however, restrictions apply to the availability of these data, so they are not publicly available. The data are, however, available from the authors upon reasonable request and with permission from the National Institute for Medical Research, Tanzania.

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
