# Peer review of "Silent Tears of Midwives: ‘I Want Every Mother Who Gives Birth to Have Her Baby Alive’—A Narrative Inquiry of Midwives Experiences of Very Early Neonatal Death from Tanzania"

_children, 2023, doi:10.3390/children10040705_

Round 1

Reviewer 1 Report

I am pleased to have the possibility to review the study “Silent Tears of Midwives: ‘I Want Every Mother Who Gives Birth to Have Her Baby Alive’. A Narrative Inquiry of Midwives Experiences of Very Early Neonatal Death from Tanzania”. The discussed problem of fetal death in Africa has a meaningful impact on maternal and children’s health care system and has meaningful long-term consequences. Moreover, the midwife's relation to these events is undoubtedly underestimated as having a meaningful impact on healthcare quality. The size of the study population and 12 weeks of follow-up are undoubtful study strengths. This study is very interesting and should be published. Nevertheless, several methodological issues require clarification before publication. 

Major points:

1.     The objectivization of results could provide a better conclusion for the study. For example, a validated questionnaire to assess PTSD or burnout symptoms by midwives.

2.     The interpretation of the result is difficult to interpret. Nevertheless, it provides the reader with very interesting information.

3.     The themes shown in the results section should be better explained in the methodology. Why were these points chosen, and what do the authors want to conclude by examination of the midwives in those categories?  

4.     The abstract is written a little bit chaotic, and the structure of the study is not seen in this part of the study. As well as, the aim and conclusions of the study are not well underlined. I will strongly recommend making these two points clearer in the abstract.

5.     The statistic or descriptive information about the count of stillbirths / death births / death after birth at the time of the inclusion period per midwives is necessary to show the influence of this complication on several midwives included. If midwives who were included in the study had not had any death birth in the period of inclusion, then an examination of their emotions is less informative. 

6.     Discussion subtitle 4.1 should be included in the methodology or result section. Please follow the guidelines for narrative reviews or clinical studies. COREQ32 checklist criteria should be included as a supplementary file.

7.     The discussion section is long, but the clinical usefulness of the study was not underlined. I suggest discussing the possible usefulness of the presented information in. the last paragraph before the conclusion. Also, this implication should be possible to perform in Tanzania. 

8.     The first paragraph of conclusion is not the conclusion of the study “In low-income countries, midwives are at the forefront of the phenomenon of VEND and may provide insight for potential solutions. Solutions for VEND are not simple; VEND remains a complex interplay between socioeconomic factors, maternal education and health, and tribal and cultural traditions, with the management of safe labour and birth beginning many months before with prenatal and antenatal care and managing labour and delivery with attentiveness. A multidisciplinary approach addressing access to higher care, available transport, and timely actions in light of obstetric complications all contribute positively to reducing VEND.” – please make this section more essential with the most important information of the study.

9.     „Every midwife is worthy of being heard; we must listen to their stories to support 430 our colleagues in their grief [105].”- conclusion of the study should be made by authors and not cited from other studies.

Minor points:

1.     I would recommend deleting the subtitles in the introduction section. 

2.     Misspelling errors in the abstract. Line “40”, for example.

105 citations - is too much for this type of study. I will recommend reducing this number and focusing on the most relevant related studies by including more interpretation of results from the authors themself.

Author Response

1.The objectivization of results could provide a better conclusion for the study.

For example, a validated questionnaire to assess PTSD or burnout symptoms by midwives.

We did consider this type of validation – but it was felt that perhaps a survey would detract from the essence of the research – the ‘voices and stories directly from the midwives. There is a dearth of face-to-face interviews with midwives in LIC – especially sub-Saharan Africa so it was felt that semi-structured interviews gave the midwives a place and space to tell it from the heart.

Burn-out survey’s (for example Maslach Burnout Inventory) and PTSD surveys can be brief, fast and may provide data that could validate their stories and insights.

However, the limitations we felt in this LIC setting was finding a cultural fit, many surveys appeared to lack cultural sensitivity and/or were culturally biased to HIC settings. Many were developed by American Psychological Association (Moore et al., 2020).

Further, whilst translation into the Kiswahili language was doable, many words used in the survey’s we looked at were problematic to translate into Kiswahili – (risk of being lost in translation) - the vernacular of many of these surveys appeared geared to HIC. Some questions could be taken as insensitive, or words didn’t keep their intended meaning.

For example, ‘cynicism’ translates poorly into Kiswahili - as ‘anxiety’.

We hope to further this research with focus groups and interventions being measured against wellbeing. But that is not in this study.

Cost was also a consideration as many of the proprietary survey had a per person cost per use and reports were an extra cost.

2.The interpretation of the result is difficult to interpret. Nevertheless, it provides the reader with very interesting information.

The authors are happy to incorporate any suggestions to make the results less difficult to read.

3.Themes:

The themes shown in the results section should be better explained in the methodology.

Extra explanation was added as suggested to clarify the thorough robust process and eventual final themes.

No themes were identified in advance; instead, the analysis was conducted in an active, creative, inductive manner, such that concepts were primarily derived and themes were conceptualized from the raw data categorization by a researcher. Researchers, JB, CB and RA independently identified patterns, clusters of similar meaning. All joined the robust discussion several times with final themes and quotes agreed upon to enhance rigor and further reduce bias. The analytical framework of Braun et al. (Braun et al., 2014) guided the stages of the interpretive reflexive thematic analysis process (see Table 1). Using hard copies, highlighters, and pencils initially, followed by each researcher independently placing quotes and themes into a word document for group discussion. The final themes were selected to best express the meanings presented in the participants stories (Braun & Clarke, 2021). This was not a linear process; rather, researchers returned purposefully, and repeatedly to engage critically with the data to ensure that themes captured the depth and breadth of the midwives’ narratives.

162-175

3. Themes:

Why were these points chosen, and what do the authors want to conclude by examination of the midwives in those categories?

Why were these points chosen?

The themes were derived from the data - not assigned to an already labelled theme.

It was a long, detailed, and robust process - the thematic development with a thorough back and forth process with much discussion, with each researcher always keeping the dataset in the forefront of the process.

The process of what we learned/derived/analysed from the midwives’ narratives was many collective similar narratives that created the ‘themes’ (categories).

3. Themes:

and what do the authors want to conclude by examination of the midwives in those categories?

What do the authors want to conclude by examination of the midwives in those categories?

The conclusion is to allow the voice of the midwives to shine and the resonating themes – this may allow other colleagues to find quiescence in the responses – midwives were not assigned a category – the themes transpired from the data.

I may have misunderstood your questions -But I hope I have addressed your concerns and comments.

4.The abstract is written a little bit chaotic, and the structure of the study is not seen in this part of the study. As well as, the aim and conclusions of the study are not well underlined. I will strongly recommend making these two points clearer in the abstract.

The aim and conclusion were reworked as per your suggestions.

Background: Midwives working in settings with limited clinical resources experience high rates of very early neonatal deaths. Midwives manage the impact of this grief and trauma almost daily, which may affect patient care and their own well-being.

Research Aim: To explore how midwives cope with high rates of very early neonatal deaths and gain midwives’ insights and solutions to reduce very early neonatal death in limited resource settings. Additionally, listen to the stories of midwives to create global collective awareness and support for midwives and their critical work in low resource settings.

Methods: Narrative inquiry utilizing semi structured interviews. Twenty-one midwives with at least six months experience who had experienced or witnessed very early neonatal death were interviewed. Data were audio recorded and transcribed, and reflexive thematic analysis of transcripts was conducted.

Results and Discussion: Three themes were identified: 1) deep sadness due to dead babies leading to internal struggle; 2) spirituality, including the use of prayer and occasional beliefs that unexplainable deaths were ‘god’s plan’; and 3) the development of resilience by seeking solutions, educating themselves, taking accountability and guiding mothers. The midwives noted that inadequate staff and high caseloads with limited basic supplies hindered their clinical practice. Midwives articulated that they concentrated on active solutions to save babies during labour, such as vigilant foetal rate heart monitoring and partogram. Further, reduction and prevention of very early neonatal death is a complex problem requiring multidisciplinary teams and woman-centred care approaches to address issues contributing to the health of mothers and their newborns.

Conclusion: Midwives’ narratives highlighted insights into ways of coping with grief and deep sadness, through prayer, and further education of both mothers and fellow colleagues to achieve better antenatal and intrapartum care and outcomes. This study gave midwives an opportunity for their voices to be heard with solutions and insights being shared with other colleagues globally in similar settings, which may offer support and encourage resilience.

25-50

5.the statistic or descriptive information about the count of stillbirths / death births / death after birth at the time of the inclusion period per midwives is necessary to show the influence of this complication on several midwives included.

If midwives who were included in the study had not had any death birth in the period of inclusion, then an examination of their emotions is less informative.

I can understand your comments but must assure you in this setting the impact of VEND, FSB’s and MSB is experienced at the least weekly.

Statistic provided by Amana were provided by the Maternal and Neonatal death causes meeting - for some of the time frame for the study. They are not approved to be published.

Neonatal Mortality rate in Labour ward and Neonatal Unit

Dates: January – June 2019

256 Deaths

(Includes 22 Intrauterine fetal death (IUFD), Fresh still birth & Macerated Still Birth).

·             206 (Labour ward)

·             50 (Neonatal Unit)

·             All staff in the study confirmed they have experienced VEND – and many stated they had seen many every week.

Excluding the IUFD neonatal deaths in labour ward is on average one neonate dying per day.

Further the ward is open plan with mothers less than one metre away from each other and staff shortages meant midwives were involved in or witnessing VEND often.

6.Discussion subtitle 4.1:  should be included in the methodology or result section.

As suggested this paragraph was moved to Methodology section.

4.1. Rigor

Participants of different ages and with different levels of clinical experience were sought to optimize the representation of midwives in this study as well as to confirm its credibility and support its transferability (Johnson et al., 2020). Rigor was supported by the COREQ32 checklist criteria (Tong et al., 2007). Narrative inquiry was best suited to elicit rich stories from midwives, whose voices have been rarely heard. An interview guide was utilized, and interviews were conducted by the same researcher to enhance reliability (Clandinin, 2016).

To address the potential influence of preconceptions and bias, a collaborative acknowledgement was made by highlighting the clinical experiences at the research site alongside perceived biases and assumptions by the researchers JB, CB, RA, MS and JM. Prolonged engagement and “fitting in” by JB, CB and JM were achieved by immersion in the cultural and clinical context for more than 7 years, which supported reflexivity and encouraged participants to share their stories more openly (Patton, 1999).

All authors contributed to the diversity of the research in terms of professional and academic skills, which enhanced the final reflexive thematic considerations of this research and thus its trustworthiness (Morse, 2015).

The research group conducted peer review sessions to ensure the trustworthiness of the midwives’ stories (Morse, 2015).

179-196

6. Discussion:

Please follow the guidelines for narrative reviews or clinical studies.

COREQ32 checklist criteria should be included as a supplementary file.

COREQ32 Checklist was utilised for this study.

The checklist is now included as a Supplementary file as per suggestion.

7.Discussion:

The discussion section is long, but the clinical usefulness of the study was not underlined.

I suggest discussing the possible usefulness of the presented information in. the last paragraph before the conclusion. Also, this implication should be possible to perform in Tanzania. 

The insights and solutions conveyed by the midwives can have clinical impact and usefulness simply by allowing the stories to be heard, opening dialogues with management to adopt a team problem solving approach for example, streamline labour ward caseloads, utilization of partograms and support neonatal resuscitation training.

381-384

8.Conclusion:

The first paragraph of conclusion is not the conclusion of the study “In low-income countries, midwives are at the forefront of the phenomenon of VEND and may provide insight for potential solutions. Solutions for VEND are not simple; VEND remains a complex interplay between socioeconomic factors, maternal education and health, and tribal and cultural traditions, with the management of safe labour and birth beginning many months before with prenatal and antenatal care and managing labour and delivery with attentiveness. A multidisciplinary approach addressing access to higher care, available transport, and timely actions in light of obstetric complications all contribute positively to reducing VEND.” –

Please make this section more essential with the most important information of the study.

Thank you for your insight and suggestion.

The paragraph was moved earlier in the discussion section.

Solutions for VEND are not simple; VEND remains a complex interplay between socioeconomic factors, maternal education and health, and tribal and cultural traditions, with the management of safe labour and birth beginning many months before with prenatal and antenatal care and managing labour and delivery with attentiveness. A multidisciplinary approach addressing access to higher care, available transport, and timely actions in light of obstetric complications all contribute positively to reducing VEND.

Additional words were added:

Research can be a conduit to validate the essence of the midwives’ day-to-day struggles in low resource areas and provide them and their colleagues with a sense of solidarity and hope.

405-412

9.„Every midwife is worthy of being heard; we must listen to their stories to support 430 our colleagues in their grief [105].”- conclusion of the study should be made by authors and not cited from other studies.

Thank you for your comments. We have removed the citation, so the statement stands alone.

Every midwife is worthy of being heard; we must listen to their stories to support our colleagues in their grief.

446-447

Minor points

Minor points:

1.I would recommend deleting the subtitles in the introduction section. 

Some subtitles were removed as suggested.

Research aim remains for clarity.

2.Misspelling errors in the abstract. Line “40”, for example.

Spelling checked by authors plus through AJE editing serviced.

3. 05 citations - is too much for this type of study. I will recommend reducing this number and focusing on the most relevant related studies by including more interpretation of results from the authors themself.

Citations reduced by 30 as per suggestion from 105 down to 75.

484-652

References:

Braun, V., & Clarke, V. (2021). One size fits all? What counts as quality practice in (reflexive) thematic analysis? Qual. Res. Psychol., 18(3), 328–352. https://doi.org/10.1080/14780887.2020.1769238

Braun, V., Clarke, V., & Rance, N. (2014). How to use thematic analysis with interview data (process research). In N. P. Moller & A. Vossler (Eds.), The Counselling & Psychotherapy Research Handbook (pp. 183–197). Sage.

Clandinin, D. J. (2016). Narrative inquiry: A methodology for studying lived experience. Res. Stud. Music Educ., 27(1), 44–54. https://doi.org/10.1177/1321103x060270010301

Johnson, J. L., Adkins, D., & Chauvin, S. (2020). A review of the quality indicators of rigor in qualitative research. Am. J. Pharm. Educ., 84(1), 7120. https://doi.org/10.5688/ajpe7120

Moore, A., van Loenhout, J. A. F., de Almeida, M. M., Smith, P., & Guha-Sapir, D. (2020). Measuring mental health burden in humanitarian settings: a critical review of assessment tools. Glob Health Action, 13(1), 1783957. https://doi.org/10.1080/16549716.2020.1783957

Morse, J. M. (2015). Critical analysis of strategies for determining rigor in qualitative inquiry. Qual. Health Res., 25(9), 1212–1222. https://doi.org/10.1177/1049732315588501

Patton, M. Q. (1999). Enhancing the quality and credibility of qualitative analysis. Health Serv. Res., 34(5 Pt 2), 1189–1208. https://www.ncbi.nlm.nih.gov/pubmed/10591279

Tong, A., Sainsbury, P., & Craig, J. (2007). Consolidated criteria for reporting qualitative research (COREQ): A 32-item checklist for interviews and focus groups. Int. J. Qual. Health Care, 19(6), 349–357. https://doi.org/10.1093/intqhc/mzm042

Please See Attachment of Table - unsure if formating would be clear in this submission form

Reviewer 2 Report

Generally, this is a powerful piece bringing the oft-unheard voice of the midwife to the forefront on a very important topic. Kudos to the authors for understanding the importance of this story. 

Regarding the abstract (lines 25-45): It's unclear if it is a formatting issue, but as it stands there is no heading to each section of the abstract to clearly identify what it is. For example, in line 25 for it to say "Background: Midwives working in settings with limited clinical resources..." And similarly for the additional abstract sections. 

For the "Keywords" lines 46-47: consider adding additional keywords including things like "global health" and "low-income settings".

Lines 87-92 are great and some of the most powerful in the manuscript!

In line 287, the acronym "HICs" does not need to be introduced again as it was early in the manuscript in line 61. 

In line 359, the acronym "HOT" for "hands-on-care" is not needed since it only appears once in the manuscript. 

Author Response

Please see attachment - as unsure if formating of response table would carry over.

Reviewer Number

Authors Response

Line

Reviewer 2 Comments and Suggestions for Authors

Generally, this is a powerful piece bringing the oft-unheard voice of the midwife to the forefront on a very important topic.

Kudos to the authors for understanding the importance of this story. 

Many Thanks for taking the time to review this manuscript and we certainly agree with you. Such an under researched region and population.

Regarding the abstract (lines 25-45): It's unclear if it is a formatting issue, but as it stands there is no heading to each section of the abstract to clearly identify what it is.

For example, in line 25 for it to say "Background: Midwives working in settings with limited clinical resources..." And similarly for the additional abstract sections. 

Formatting issue resolved and headings now added for better flow and clarification.

The aim and conclusion were reworked as per your suggestions.

Background: Midwives working in settings with limited clinical resources experience high rates of very early neonatal deaths. Midwives manage the impact of this grief and trauma almost daily, which may affect patient care and their own well-being.

Research Aim: To explore how midwives cope with high rates of very early neonatal deaths and gain midwives’ insights and solutions to reduce very early neonatal death in limited resource settings. Additionally, listen to the stories of midwives to create global collective awareness and support for midwives and their critical work in low resource settings.

Methods: Narrative inquiry utilizing semi structured interviews. Twenty-one midwives with at least six months experience who had experienced or witnessed very early neonatal death were interviewed. Data were audio recorded and transcribed, and reflexive thematic analysis of transcripts was conducted.

Results and Discussion: Three themes were identified: 1) deep sadness due to dead babies leading to internal struggle; 2) spirituality, including the use of prayer and occasional beliefs that unexplainable deaths were ‘god’s plan’; and 3) the development of resilience by seeking solutions, educating themselves, taking accountability and guiding mothers. The midwives noted that inadequate staff and high caseloads with limited basic supplies hindered their clinical practice. Midwives articulated that they concentrated on active solutions to save babies during labour, such as vigilant foetal rate heart monitoring and partogram. Further, reduction and prevention of very early neonatal death is a complex problem requiring multidisciplinary teams and woman-centred care approaches to address issues contributing to the health of mothers and their newborns.

Conclusion: Midwives’ narratives highlighted insights into ways of coping with grief and deep sadness, through prayer, and further education of both mothers and fellow colleagues to achieve better antenatal and intrapartum care and outcomes. This study gave midwives an opportunity for their voices to be heard with solutions and insights being shared with other colleagues globally in similar settings, which may offer support and encourage resilience.

25-50

For the "Keywords" lines 46-47: consider adding additional keywords including things like "global health" and "low-income settings".

Good suggestion – Key words added.

·       Global Health

·       low-income settings

Lines 87-92 are great and some of the most powerful in the manuscript!

Thank you for your comments.

In line 287, the acronym "HICs" does not need to be introduced again as it was early in the manuscript in line 61. 

HIC removed as suggested

In line 359, the acronym "HOT" for "hands-on-care" is not needed since it only appears once in the manuscript. 

HOT removed as suggested

Round 2

Reviewer 1 Report

This type of study is very difficult to perform and presenting the results to make them easy to read is an even harder challenge.

Thank you very much for providing me with such detailed information with your answer. Nevertheless, more important is to include explanations in your manuscript to make it more clear and more transparent about what manly was performed. I have only several minor suggestions.

- I will recommend including your answers to my questions according to "3. Themes" into your manuscript.

- W will recommend adding the information "The insights and solutions conveyed by the midwives can have clinical impact and usefulness simply by allowing the stories to be heard, opening dialogues with management to adopt a team problem solving approach for example, streamline labour ward caseloads, utilization of partograms and support neonatal resuscitation training." to your discussion if it is missing.

- I will recommend adding the folowing information to your manuscript:

"Dates: January – June 2019  256 Deaths  (Includes 22 Intrauterine fetal death (IUFD), Fresh still birth & Macerated Still Birth).  ·             206 (Labour ward)  ·             50 (Neonatal Unit)  ·             All staff in the study confirmed they have experienced VEND – and many stated they had seen many every week.  Excluding the IUFD neonatal deaths in labour ward is on average one neonate dying per day.  Further the ward is open plan with mothers less than one metre away from each other and staff shortages meant midwives were involved in or witnessing VEND often"

Author Response

Reviewer Comments |Notes

Authors Response

Line Number

This type of study is very difficult to perform and presenting the results to make them easy to read is an even harder challenge.

Thank you very much for providing me with such detailed information with your answer. Nevertheless, more important is to include explanations in your manuscript to make it more clear and more transparent about what manly was performed. I have only several minor suggestions.

Thank you for your time and expediency in which you have reviewed this manuscript. It is greatly appreciated.

As suggested, we have incorporated the explanations into the manuscript.

- I will recommend including your answers to my questions according to "3. Themes" into your manuscript.

Thankyou for your review. As suggested the answer from Reviewer table one has been incorporated into the data analysis.

The themes were derived from the data - not assigned to an already labelled theme. The analysis was a long, detailed, and robust process - the thematic development with a thorough back and forth process with much discussion, with each researcher always keeping the dataset in the forefront of the process.

The process of what we learned, derived, and analysed from the midwives’ narratives focused on similar narratives that created the ‘themes’ (categories).

The themes identified focus attention on the experiences of midwives and the need to address sadness resulting from VENDS via spiritual practices/beliefs and solution generation/adoption.

356-364

Tracked Changes

- W will recommend adding the information "The insights and solutions conveyed by the midwives can have clinical impact and usefulness simply by allowing the stories to be heard, opening dialogues with management to adopt a team problem solving approach for example, streamline labour ward caseloads, utilization of partograms and support neonatal resuscitation training." to your discussion if it is missing.

Thank you for your review. As suggested the information has been added to the discussion.

The insights and solutions conveyed by the midwives can have clinical impact and usefulness simply by allowing the stories to be heard, opening dialogues with management to adopt a team problem solving approach for example, streamline labour ward caseloads, utilization of partograms and support neonatal resuscitation training

574-577

Tracked Changes

- I will recommend adding the folowing information to your manuscript:

"Dates: January – June 2019  256 Deaths  (Includes 22 Intrauterine fetal death (IUFD), Fresh still birth & Macerated Still Birth).  ·             206 (Labour ward)  ·             50 (Neonatal Unit)  ·             All staff in the study confirmed they have experienced VEND – and many stated they had seen many every week.  Excluding the IUFD neonatal deaths in labour ward is on average one neonate dying per day.  Further the ward is open plan with mothers less than one metre away from each other and staff shortages meant midwives were involved in or witnessing VEND often"

Thank you for your review. As suggested the information has been added to the manuscript.

Dates: January – June 2019 saw 256 Neonatal Deaths  (Includes 22 Intrauterine Fetal Death (IUFD), Fresh Still Birth & Macerated Still Birth). 206 (Labour ward) and 50 (Neonatal Unit).

All staff in the study confirmed they have experienced VEND – and many stated they had seen many every week. Excluding the IUFD neonatal deaths in labour ward is on average one neonate dying per day. Further the ward is open plan with mothers less than one metre away from each other and staff shortages meant midwives were involved in or witnessing VEND often.

240-247

Tracked Changes

Additional Grammar and edits were attended to.

See Track Changes.

See Attached File
